# The Professional and Psycho-Emotional Impact of the COVID-19 Pandemic on Medical Care—A Romanian GPs’ Perspective

**DOI:** 10.3390/ijerph18042031

**Published:** 2021-02-19

**Authors:** Celina Silvia Stafie, Lenuta Profire, Maria Manuela Apostol, Irina Iuliana Costache

**Affiliations:** 1Department of Preventive Medicine and Interdisciplinarity, Faculty of General Medicine, “Grigore T. Popa” University of Medicine and Pharmacy of Iasi, 16 Universității Street, 700115 Iasi, Romania; 2Department of Pharmaceutical Sciences I, Faculty of Pharmacy, “Grigore T. Popa” University of Medicine and Pharmacy of Iasi, 16 Universității Street, 700115 Iasi, Romania; 3Prolife Clinics, 28 Anastasie Panu Street, 700115 Iasi, Romania; apostol.manuela@yahoo.com; 4Department of Medical Sciences I, Faculty of General Medicine, “Grigore T. Popa” University of Medicine and Pharmacy of Iasi, 16 Universității Street, 700115 Iasi, Romania; irina.costache@umfiasi.ro

**Keywords:** COVID-19, Romanian GPs, stress, adjustment to pandemic, burden of prevention

## Abstract

The aim of this study was to assess the psycho-emotional impact and the adjustment degree of Romanian general practitioners (GPs) in the coronavirus disease 2019 (COVID-19) pandemic context. With a cross-sectional design, the study included 677 GPs to whom a validated questionnaire based on different items targeting three factors was sent: burden of prevention, presence of stress symptoms, and adjustment to pandemic. The burden of prevention and the adjustment effort to the pandemic were felt significantly more by female doctors and by GPs working in associated offices. The case definition quality, the support received, the professional life changes, and the stress symptoms proved to be the main predictors for the adjustment to pandemic. The adjustment measurement questionnaire can be used in further studies to identify the most supportive public health practices in difficult epidemiological contexts.

## 1. Introduction

On 11 March 2020, the World Health Organization (WHO) declared the outbreak caused by the new coronavirus disease 2019 (COVID-19) an international pandemic. In Romania, the first infected patient was diagnosed on February 26, but the authorities established the state of emergency on March 16. Under the time pressure, doctors had to deal with different simultaneous challenges: (i) preventing the virus transmission; (ii) protecting the medical staff and teaching ways to prevent the spread of the virus; (iii) adapting the management and providing the necessary materials for the specific COVID-19 medical services; and (iv) preserving its state of health and well-being [1,2].

The daily pressure of general practice can be understood through a combination of factors. There are three main categories involved, the patient factors (e.g., the desire for both rapid access and continuity of care, the treatment expectations, the patient complexity, and polypharmacy or repeat prescribing), the system factors, like introduction of new services, new medication (an increasing use of medications for primary and secondary prevention, such as statins, new antihypertensives and new anticoagulants, to be prescribed and reviewed), or new preventive services, the relationships with the wider system (mental health services, social care services and the third sector), and the supply-side factors (funding, workforce migration and instability) [3]. The fourth factor is very finely insinuated into our existence, being embedded in it, which makes it difficult to be perceived separately—the pressure of social media and mass media [4,5].

For patient’s desire to rapid access and continuity of care, in order to release the pressure on general practitioners (GPs), methods to manage demand have been introduced, particularly for managing same-day presentation of acute onset illness (telephone triage schemes and changing skill-mix by using other members of the primary care team) [6].

The system factors involved in the burden are more difficult to solve. The rule of any new and imposed change, such as new preventive services for GPs, is to develop contradictory feelings and opposition. Therefore, the targeted population, the GPs, have perceived an enhanced burden of prevention, especially while facing pandemic [3].

The past ten years imposed an additional workload, with numerous vaccines in the immunization program, including extending influenza vaccines to young children and pregnant women and the introduction of rotavirus, meningococcal B vaccine, all administered in primary care. This year, 2021, imposed the newest vaccine, anti-COVID-19, only that side effects and post-vaccine information is missing or changing every day. This is an extreme and unpredictable factor for additional pressure for GPs [3,7].

Speaking about new preventive services, now, the preventive work is a core part of the role of primary care [3]. GPs are monitored and incentivized to undertake large amounts of disease monitoring and prevention. In all this context of preexistent unsolved increasing burden of general practice, 2020 brought us, globally, to the maximum capacity of our preventive expertise, creating a new meaning for the term “burden of prevention”.

Some of the challenges that were reported by the GPs in this pandemic were similar to those earlier reported in evaluations of primary care response to previous health crises [8], such as influenza outbreaks: “lack of support from authorities, burden of prevention due to the lack of personal protective equipment (PPE), additional professional tasks and the emotional burden” [9].

In several countries, the demand for medical care during the COVID-19 pandemic exceeded the available resources, urging stakeholders to reorganize the medical landscape. The health care systems became overwhelmed and placed the heaviest burden on two health care providers, specialized units in hospitals and family doctors [10]. Primary care literature on the impact of the COVID-19 outbreak on primary care is still emerging. A brief summary of the main international registries and European national registries, designed to search for planned or on-going COVID-19 clinical trials, are reuniting WHO International Clinical Trials Registry Platform (ICTRP), ClinicalTrials.Gov, EU Clinical Trials Registry, and Cochrane Central Register of Controlled Trials (CENTRAL) [6]. All these COVID-19 clinical trials are focused on medical research for treating COVID-19, while few studies are focused on medical staff and social workers [11,12].

There was a common lack of information about COVID-19, between Romania and other countries, such as the UK [13], the United States, Germany, or the Netherlands [14]. According to the European Centre for Disease Prevention and Control (ECDC), there were also tremendous differences regarding the testing rate. The understanding of the evolution of the COVID-19 pandemic is built upon data concerning confirmed cases and deaths. This data can be meaningfully interpreted only alongside with the virus data testing in different countries [15]. A comparative look at the testing rate of week 45 (the second week of November 2020), shows that a higher rate was present in severely affected countries like Italy, 2176, or France, with 2359 [16], as well as some of the less affected countries, like the Czech Republic, which had a testing rate of 2527. Even though Greece had a similar early lockdown reaction, like the Czech Republic, we noticed a low testing rate, of only 1173 [15]. One of the lowest rates was in Romania, with only 1053 [16].

The countries located on the border with Romania showed different prevalence in testing: Hungary, 1395; Bulgaria, 1018; while Serbia, Moldova, Ukraine [17], and Northern Macedonia did not declare their rate testing at all. The last four countries referred only to the total number of cases in the second week of 2021 and failed to declare the test rate [15,16].

Almost all affected countries looked with great suspicion at the virulence and speed potential of the new Severe Acute Respiratory Syndrome Coronavirus 2 (SARS-CoV-2), concerned about people’s well-being [18]. In Romania, the solutions adopted by the Ministry of Health were not enough to practically protect and support the doctors during the pandemic. For example, British doctors involved in the epidemic benefited from a number of facilities, such as preferential supermarket shopping programs and discount prices, well-being support services, free guides, and apps with suggestions for their mental and physical health [19]. The current study aimed at showing the real picture of a society subject to media pressure, contradictory news, and dysfunctions of the administrative apparatus, but also the real condition of doctors, with fear of professional failure and fear of illness [20]. More specifically, the objective of the study was to analyze the psycho-emotional impact and the degree of general practitioners’ (GPs’) adaptation and acceptance towards the changes of working conditions, imposed by the COVID pandemic. Under current specific conditions, family physicians have been appointed to take over outpatient tasks and perform physical or online triage in patients with COVID. We believe that an assessment of their health, safety, and satisfaction is essential for the proper functioning of the health system [21]. The study was based on a self-reported questionnaire conducted to assess the health and well-being of GPs and to investigate the impact of pre-existing pathologies on the physical and mental health of GPs during the pandemic time.

## 2. Materials and Methods

### 2.1. Study Design

The study is a social research, which aims to assess the impact of living and working conditions during the current pandemic on family doctors in Romania, related to their objective resources (e.g., endowments) and subjective resources (e.g., psychological state). It is a prospective open, cross-sectional study, in which the questionnaire method was used, collecting answers by items. There was no alternative, since study debuted in an emergency state, when only forms of online communication were permitted. The interpretation of the following aspects was followed: the perceived effort for occupational adaptation to the epidemic, the attitude towards possible risks related to one’s own health, the symptoms associated with stress and those related to the clinical picture of the disease given by the COVID-19 virus, and aspects related to the perceived instrumental and informational support and received from the authorities.

### 2.2. Data Collection

The participants in the study were family doctors from Romania, randomly selected, who agreed to answer the proposed questionnaire. The selected participants were contacted through social media professional groups, “Romanian General Practitioners on Facebook”. The request for participation and the link to the online questionnaire were sent together with the informed consent and a short letter explaining the goals of the study and asking for honest answers. In case of a positive agreement, the participants filled in the online form of this questionnaire, using Google Forms technology and their answers were automatically recorded in the database. The data were collected between 30 April and 2 May 2020, approximately two months after the state of emergency was declared in Romania.

The design and validation of the questionnaire. A questionnaire based on three factors was designed and validated to perform the study: (i) burden of prevention, (ii) presence of stress symptoms, and (iii) adjustment to pandemic, each factor containing different items. The questionnaire was elaborated by a group of experts (five GPs with thorough experience in the field and four epidemiologic doctors from the Public Health Department). Initially, 31 items had been established. The alpha-Cronbach consistency index [22] obtained for each item points to the validity of the questionnaire: burden of prevention (6 items, the alpha-Cronbach coefficient is 0.899), the presence of the stress symptoms (12 items, the alpha-Cronbach coefficient is 0.886), adjustment to pandemic (13 items, the alpha-Cronbach coefficient is 0.814). The pretesting of the questionnaire was validated on a sample of 103 subjects and we opted for a 6-point assessment scale (1—to a small extent, 6—to a large extent) [23]. After the content and construct validity pretesting, the final questionnaire comprised 29 items. Two independent assertions were added at the end of the standardized questionnaire which investigated the GPs’ perception on the definition initially elaborated by the WHO and the evaluation of the modification of a cabinet’s professional activity [24,25]. Participants: The study was carried out at the level of the whole country, as well as by comparison between geographical regions. Different inclusion and exclusion criteria were applied to enroll the participants to the study. Criteria of inclusion: assuming all GPs had to deal with COVID-19 while being active, GPs from all regions of Romania, aged between 25–65, in activity, were invited to participate to the study. Criteria of exclusion: inactive GPs at the moment of enrollment in the study, medical students, residents, retired GPs, other medical specialties, nurses, and all GPs who did not want to comment on their activity or freely share their professional experience. The results of the Kolmogorov–Smirnov test indicated that the data were normally distributed (*p* > 0.05) [23].

The Nomenclature of Territorial Units for Statistics (NUTS) presents Romania as four macro-regions, on four administrative-territorial levels: (1) Romanian counties and Bucharest (NUTS III; *k* = 42), (2) development regions (NUTS II; *k* = 8), (3) development macro-regions (NUTS I; *k* = 4), and (4) broadly defined traditional historical regions (Moldova, Transylvania, and Wallachia; *k* = 3. The equivalences of regions are illustrated in the NUTS II map, included in Appendix A [26].

The sample consisted of 677 subjects from all eight geographical regions of Romania, the west region (RO42), Bucharest-Ilfov (RO32), north-east (RO21), south-east (RO22), south-Muntenia (RO31), north-west (RO11), center region (RO12), and south-west region(RO41), which fits the number of inhabitants of these regions and the total number of GPs enlisted there [26,27] (Appendix A map NUTS II).

### 2.3. Research Ethics

All subjects signed an informed consent for inclusion before they participated in the study. The study was conducted in accordance with the Declaration of Helsinki, and the protocol was approved (27 April 2020) by the Ethics Committee of “Grigore T. Popa” University of Medicine and Pharmacy of Iasi, Romania.

### 2.4. Statistical Analysis

The data were analyzed using SPSS for Windows, Version 19.0 (SPSS Inc. Chicago, IL, USA). Pearson’s correlation, independent sample *t*-test, and stepwise multiple regression analysis were used to evaluate the results. The continuous and scale variables were presented with means and standard deviations (SD), and the categorical ones by frequencies and percentages (%). 

The dependent variables were adjustment to pandemic, burden of prevention, and stress level of doctors. The main dependent variable was the degree of adjustment to pandemic, evaluated on a scale from 1 to 12 and calculated using the subjects’ binary (yes/no) answers to 12 relevant items. Taking into consideration that the median value for this variable was equal to 9.00, we binarized it, by defining two possible states:

Lower adjustment: corresponding to all values less or equal than 9.00;

Higher adjustment: corresponding to all values greater than 9.00.

Independent variables: The independent study variables were sociodemographic characteristics (gender, age, environment, medical offices grouped/isolated), comorbidities (obesity, high blood pressure, diabetes), the presence/absence of the contact with COVID patients, knowledge level about the COVID pandemic (quality of the case definition), and professional life status (public health involvement and received support from the authorities). Most variables were scalar, being calculated according to the subjects’ binary responses to specific items. Finally, our intention was to determine the predictors for the three dependent variables (factors) of the questionnaire. Sample size estimation: The sample size required for our analysis was calculated to assess the validity of its results; in Romania, there are 12,072 general practitioners (GPs), working in private and public institutions; in order to maintain an error level smaller than 5%, a minimum sample size of 372 subjects was required. Our sample size was 677 subjects, which exceeds clearly the minimal requirement [28].

## 3. Results

This cross-sectional study is designed to evaluate Romanian GPs’ adjustment to the COVID-19 pandemic. In Romania, there are about 45,000 physicians, of whom 12,072 are GPs [27] working in private and public institutions. The study was performed approximately two months after the establishment of country measures for the state of emergency generated by the new SARS-CoV-2 coronavirus. A brief description of the principal socio-demographic characteristics of the sample, mean age ± SD, and gender differences is presented in Table 1.

Of all 677 subjects who participated in the study, 83% show different chronic disease while only 17% show no chronic disease up to the moment of inclusion in the study. Referring to the GPs with chronic diseases, 22.7% show high blood pressure (HBP) under treatment, 16.5% are obese, 5.3% suffer from diabetes, and 2.6% have chronic pulmonary conditions, asthma, or chronic obstructive pulmonary disease (COPD). All these conditions are considered an additional risk for the SARS-CoV-2 infection. There remains 52.9% of GPs included in the research lot showing other chronic diseases (autoimmune diseases, kidney disorders, recurrent depressive disorders, anxiety disorders, etc.) (Table 2).

A percentage of 6.35 come from the West region, followed by the Bucharest-Ilfov region (19.79%), North-East region (14.18%), South-East region (14.33%), South-Muntenia region (13.29%), North-West region (13.15%), Center region (10.19%), and South-West region (8.71%) (Figure 1).

The impact of different predictors, such as gender, type of practice (individual and associated practices), age, region, chronic diseases, seniority in the profession, and working environment (rural, urban) was studied for burden of prevention, the presence of stress symptoms, and the adjustment to the pandemic of GPs. Moreover, the GPs’ perception on the case definition elaborated by the WHO was also investigated. The Pearson correlation showed a positive, low intensity, and significant correlation between the presence of the stress symptoms and the adjustment to pandemic and the burden of prevention (*r* = 0.242, *p* < 0.001) of the GPs included in the study (Table 3).

The results concerning the burden of prevention, which refers to the effort of adjustment GPs feel physically and emotionally, showed that this effort was felt significantly more by female doctors, (*t*(675) = 4.17, *p* < 0.001), by doctors working in associated cabinets, (*t*(675) = 1.76, *p* < 0.048), and by doctors over 52, (*t*(675) = 2.83, *p* < 0.005). That points to the fact that, for GPs, the stress of being infected with this virus, the fear of infecting loved ones, and the daily pressure created by the observance of the restrictions imposed by the authorities (isolation from children and the enlarged family, in an attempt to avoid contamination, etc.) represented elements that triggered a psycho-emotional imbalance [29]. GPs from grouped cabinets adapted less, compared with those having their practices in isolated cabinets. The latter were less exposed, less ill, and therefore managed better the adaptation to the pandemic [30].

The presence of stress symptoms was significantly higher among females compared to males, (*t*(675) = 4.53, *p* < 0.001). Female GPs showed a lower adaptation level to the pandemic than male GPs, which can be translated into a faulty immune status compared to males due to the higher prevalence of chronic diseases, but also to a lower stress management level compared to men [31]. There are significant differences between female and male subjects, regarding the adjustment to pandemic, defined according to the literature data [32,33] as the management of the crisis and the level of adaptation to the new working conditions. The female subjects considered that their adaptation capacities to the new conditions imposed by the pandemic were significantly more solicited, (*t*(675) = 2.167, *p* < 0.031). A still high level of professional change was also presented by GPs working in forms of professional association, (*t*(675) = 3.47, *p* < 0.001). The gender difference, with a predominant impact on female GPs related to the stress adaptation response, has multiple explanations starting from women’s typology-detail oriented, with a focus on the safety of their family and that of their patients, but with a low capacity of simultaneously managing multiple tasks in both sectors of their lives, personal and professional, justified in itself by a higher number of tasks taken on compared to men. This is also supported by the sociological evaluation of the Romanian society, still deeply traditional, which leaves on women’s shoulders both household chores and professional duties, in a somehow hypocritical attempt to observe women’s right to work equality [34,35]. Female GPs from the north east and south east had a higher nervous and physical consumption compared to their male colleagues, perceiving the pandemic context as more difficult regarding stress management, the burden of prevention, and the adjustment to pandemic.

The statistical analysis from the perspective of the presence of chronic diseases showed that GPs aged 52 and above have significantly more chronic conditions compared to their peers aged 29–52, (*t*(675) = 4.66, *p* < 0.001) and so, a high risk for SARS-CoV-2 infection. There were significant differences between GPs showing diabetes and those diagnosed with asthma, namely that diabetics show significantly fewer symptoms compared to asthmatic patients, (*t*(47) = 1.88, *p* < 0.05). We would like to indicate that the subgroups of subjects were not close in numbers so that we cannot expect significant differences.

To identify the predictors for the burden of prevention, symptoms of stress and adjustment to pandemic the regression analysis was applied (Table 4, Table 5 and Table 6). 

Regression analysis is identified as the best predictive model for adapting to the pandemic burden, seniority, type of praxis, gender, region, work environment, and age. This means that those with more seniority in the profession, working in individual offices, men, from the North West region, rural areas and >52 years old, have adapted much better.

For the presence of the level of stress dimension with GPs, the regression analysis included the following predictors in the model: seniority, the type of practice, and age (Table 5). Thus, doctors with more than 21 years in practice, of the male gender, and who perform their activity in individual cabinets were evaluated with the lowest stress level.

Applying the regression analysis, we identified that the best predictive model for the adjustment to pandemic factor comprises the predictors: a GP’s gender, age, and type of praxis. Thus, we define the optimum predictive model as “a doctor who operates in isolated practice, without physical professional associations, of male gender and aged over 52”, such being the profile that managed best the pandemic crisis (Table 6).

We also studied the GPs’ perception towards the authorities in the management of the pandemic, taking into consideration geographical regions. The GPs from the north east and the north west regions considered that the authorities got involved compared to those in Bucharest-Ilfov, who considered their implication as insufficient, with a *p* < 0.05. Similarly, GPs from the south-east had a major degree of satisfaction compared to those in Bucharest-Ilfov. Other regions which were more satisfied than the Bucharest-Ilfov region are the south-west, west, and north-west. In addition, it was noticed that the north-east region had GPs significantly more satisfied than those in the south-east region, although the same GPs had significantly more COVID symptoms compared to those in the south-east region, but also compared to those in the west region. Paradoxically, the north-west is the most satisfied region concerning the intervention of the authorities compared to the north-east, although people from the latter region showed more stress symptoms compared to their colleagues from the north-west region. A somewhat paradoxical aspect is that doctors in the north-west region, but also in the south-west region are more satisfied than those in the south-east region (*p* < 0.05), although the number of cases with COVID symptomatology of those in the south-west region is higher than those in the north-west and west regions. Doctors in the north-west and south-west regions are more satisfied than those in the south-east region. Their motivation is different. Those in the north-west, being a priority Germanic population, with rigor and compliance with procedures, had better trained medical staff in the area, and self-organized for new prevention in pandemic, while those in the south-west region had a monopoly on case management, helping colleagues in the other regions, explained by the high degree of collectivism that characterizes this region [32]. Those in the south-east region waited for the intervention of the authorities, which was delayed, so their degree of satisfaction was lower. There are significant differences between the GPs in Romania aged 29–52 regarding the perception towards the case definition, namely that they believe to a higher extent that the case definition was less clear compared to their colleagues aged 52 and above, (*t*(675) = 2.08, *p* < 0.038). There were no significant differences between males and females regarding the case definition elaborated by the WHO, both categories considering the definition as unclear and the authorities overwhelmed by the situation, *p* > 0.05. 

## 4. Discussion

According to the European Society of Traumatic Stress, the latest pandemic showed that during and after the pandemic, one in four Europeans were affected by post-traumatic stress [33]. Six months after the first global lockdown, it became obvious that 2020 made its debut with the most virulent pandemic, when the entire world was unplugged. The stress level should therefore exceed any other level that has been attested so far. According to the National Center of Surveillance and Control of Communicable Diseases, the National Institute of Public Health Bucharest, “the risk of severe condition associated to COVID-19 for people from EU/EEA and UK is deemed moderate for the general population and high for the elders and individuals with preexisting medical conditions” [2]. A similar situation is being reported in the WHO reports from 2020: “chronic diseases are an additional risk factor in contracting the virus and also in developing severe forms of SARS-CoV-2” [29].

Regarding the burden of prevention dimension, we notice that in the predictive model we have the following predictors: subjects’ gender, age, type of practice, region, and the presence of chronic diseases. An infected/ill GP, therefore unavailable for the patients, would lead to a disastrous chain of consequences regarding the supply of services for such patients, creating a major imbalance regarding the patients’ access to medical services [31]. The absence of GPs in the rural environment would create even more dramatic effects for the access to medical services of the insured.

Referring to the GPs’ perception on the COVID-19 case definition, it is important to note that WHO has changed it several times since March 2020, the latest being on 7 August 2020. Still, “the approach to defining possible and probable cases shows considerable heterogeneity” as well as “many member states still do not have an official definition of death due to COVID-19, available online” and finally “harmonization of COVID-19 case definitions is essential” [36].

According to David D, in “Research report: The psychological and psychocultural profile of Romania’s regions”, the author states that there are differences that might be considered as ecologically relevant (e.g., trust, rational beliefs, positive emotions), especially when comparing the regions across PESH indicators (political/economic/social/health). The distribution of psychological characteristics transposed over the geographical ones (geographically derived psychological maps) is different. There are six main features for the Romanians, three of them having great impact on how people react to stress and social pressure [37].

The first is “social power”, meaning that power is distributed among individuals in various areas, who control each other, to prevent the concentration of power and authoritarianism. This feature is highly manifested in NE and SE regions), described as poor regions. On the other hand, another feature studied is “avoiding/engaging uncertainty”. Avoiding uncertainty refers to the unpredictability of the future being seen as a danger, hence the current situation and Romania has a very high score at this level [32]. Almost all counties from the South-West, south, and SE regions have high scores for avoiding uncertainty, thus, always expecting others to decide, to provide for them, instead of taking action [36,37].

Another feature, important in understanding different Romanian reactions to stress, is “collectivism/versus autonomy or individualism”. Psychological collectivism is when the interests of the group are more important than those of the individual, and groups are formed in the more general logic of family relationships (e.g., family/acquaintances/friends). Romania has a high score of collectivism and, of the European Union countries, only Bulgaria, Croatia, Greece, Portugal, and Slovenia still have such a profile. The map from page 9 of the report quoted presents the autonomous counties that are concentrated in the south-west and north-west of Romania. This study results point exactly the same, that north-west and south-west regions have been more autonomous and content about their job [38].

About the region Bucharest-Ilfov, it is remarkable, in a negative way, that no feature of all six described is manifested in a personal, unique way, even if, historically, this region, including the capital, was and still remains the region that is most gifted, most targeted, most assisted, hence, with a self-perception of unique child, always deserving and gaining the best treatment. This explains the passive attitude and a certain permanent low degree of satisfaction, despite the reality.

In November 2020, there were 37 studies in the current medical literature on the impact on physical and mental health of the active population in the COVID-19 pandemic [39], the results of which highlight the differences between the effects of the pandemic on the population, based on age and level of education [40,41]. The review, that focused on measuring the effects of COVID-19 on wellness of healthcare providers between 27 April and 6 May 2020, revealed consistent reports of stress, anxiety, and depressive symptoms in healthcare workers as a result of COVID-19 [42], like all studies that reassessed the urging burden released by the pandemic, creating psychiatric damage, insomnia, depression, and anxiety among most of the healthcare workers in general practice [43].

The results obtained in our study confirm that in Romania, the stress of accommodation to the pandemic affected early physical and mental health of family doctors, a conclusion drawn six months in advance of the studies cited above. We consider it a feature of originality.

There are few studies ongoing in Romania about this subject, the psycho-emotional and professional impact of COVID-19 on health workers, and fewer are published, regarding general practitioners, that addressed the level of adaptation of doctors in the first two months of outbreak. The study we conducted has previewed the damage and taken the initiative to show the authorities that they have to improve the support and logistics and maybe to rethink the concept of well-being for their main healthcare providers, the family doctors [39,44].

The originality of the study results also from other aspects: 1. The study analyzed the hot answers obtained during the most difficult period of the pandemic, the onset, when there were almost no means of protection, and the training of family doctors was lacking, when they were forced to adapt; 2. the study compares the responses and reactions to the pandemic of doctors in different regions of the country, in an initial phase, when no one spent time for sociological research, but only for clinical observation and therapeutic intervention; and 3. it is an instant photograph of the first reactions captured among family doctors, thus constituting a faithful reproduction of the way the Romanian medical society reacted in the first two months. 

This research is not without limitations. To begin with, our survey was conducted at a single point near at the beginning of the first wave of the pandemic. It would be ideal to learn how people’s answers change as the pandemic enters the second or third wave. In terms of methodology, the web-based design means that people with no internet access and limited computer literacy were not surveyed, which explains the low percentage of respondents aged 60 or above.

Another important limitation is that there is no real homogeneity in the official declaration and registration of the number of new COVID-19 cases, registered worldwide. This uncertainty in the accuracy of the information officially provided leads to the possibility of misinterpretations of the real differences between countries, in terms of comparisons between EU countries on the pace of patient care, overwork of doctors, and the burden of prevention [14]. It should be noted that there are limitations to this type of data, including that definitions vary and the data collection process requires constant adaptation to avoid interrupted time series [45]. Testing policies and the number of tests performed per 100,000 persons vary markedly across the EU/EEA and presumably even more so among third countries [46]. More extensive testing will inevitably lead to more cases being detected. Present policy in the data report only refers to the total number of cases and deaths and to the last 14 days of new cases notifications. Total number of cases within the second week of 2021 are: Bulgaria with 214,817, Czechia 940,004, Hungary 360,418, Moldova 156,202, North Macedonia 90,654, Poland 1,478,119, Romania 712,561, Serbia 383,603, and Ukraine 1,194,328 [17], all referred to 100,000 population. The proportion between new cases should be maintained also in data testing, if reported [46].

The study, analyzing only the adaptation stress of family doctors, could benefit from increased attention if we introduce other categories of specialists, making comparisons between them [47].

Finally, the fact that it was for the first time in history, when a pandemic of such geographical dimensions was experienced simultaneously on the Internet, making information from around the world spread instantly, the accuracy of the answers could be distorted, not forgetting that panic spreads fastest with the help of social networks. The fact that the questionnaires were administered exclusively online may ease the time of data collection, but there is also a chance of it distorting the accuracy of the answers (too short, incomplete answers).

## 5. Conclusions

This paper draws attention to the fact that GPs, burdened with extra tasks during the pandemic, need psychological support in addition logistic support, as the stress factor has always been important, given that no one knows when the pandemic will end.

The GPs included in the study are considered to have received inefficient and insufficient support from the authorities, especially GPs with preexisting chronic diseases who have a diminished work capacity and endurance. This is an alarm signal for the epidemiologists and doctors in the labor medicine who need to conduct periodic controls, record their findings and issue an “able to work” or “conditioned ability to work” certificate, depending on the GP’s physical and psychic health condition.

The adjustment measurement questionnaire can be used in further studies to identify the most supportive public health practices in difficult epidemiological contexts. It is in our plans to apply it every six months, since we expect that a generalized burnout is to be expected. The protection as well as the well-being of the employees in the health field should become a national priority in the context of the COVID-19 pandemic.

## Figures and Tables

**Figure 1 ijerph-18-02031-f001:**
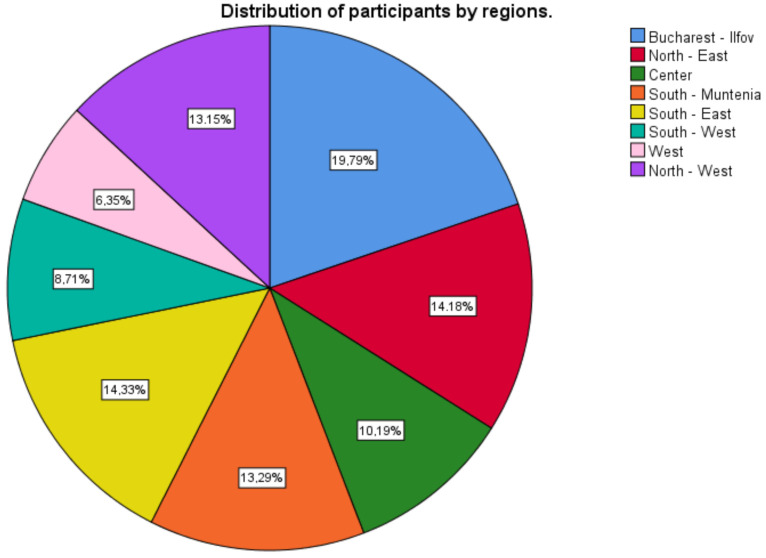
Distribution of participants by regions of Romania.

**Table 1 ijerph-18-02031-t001:** Description of socio-demographic characteristics.

Description of Socio- Demographic Characteristics	Gender	Type of Practice	Seniority in the Profession	Age	Region
Valid	677	677	677	677	677
Missing	0	0	0	0	0
Mean	1.8715	1.5569	21.9143	51.5790	4.05
Std. Deviation	0.33490	0.49712	10.83599	8.87662	2.371

**Table 2 ijerph-18-02031-t002:** Distribution of the chronic disease type of the subjects enrolled in the study.

Stage of Disease	Frequency	Percent (%)	Valid %	Cumulative %
HBP	154	22.7	22.7	22.7
COPD	5	0.7	0.7	23.5
Asthma	13	1.9	1.9	25.4
Obesity	112	16.5	16.5	41.9
Diabetes	36	5.3	5.3	47.3
Chronic hepatitis	5	0.7	0.7	48.0
Kidney disease	5	0.7	0.7	48.7
Depression	9	1.3	1.3	50.1
Anxiety syndrome	18	2.7	2.7	52.7
Panic attacks	5	0.7	0.7	53.5
Surgical intervention ^1^	5	0.7	0.7	54.2
Cancer	10	1.5	1.5	55.7
Without chronic diseases	115	17.0	17.0	72.7
Other chronic diseases	185	27.3	27.3	100.0
Total	677	100.0	100.0	

HBP = high blood pressure; COPD = chronic obstructive pulmonary disease; ^1^ = surgical intervention 3 months ago

**Table 3 ijerph-18-02031-t003:** Correlations between dependent variables.

Correlations		Adjustment to Pandemic	Presence of Stress Symptoms	Burden of Prevention
Adjustment to pandemic	Pearson correlation	1	0.242 **	0.242 **
	Sig. (2-tailed)		0.000	0.000
	*n*	677	677	677
Presence of stress symptoms	Pearson correlation	0.242 **	1	1.000 **
	Sig. (2-tailed)	0.000		0.000
	*n*	677	677	677
Burden of prevention	Pearson correlation	0.242 **	1.000 **	1
	Sig. (2-tailed)	0.000	0.000	
	*n*	677	677	677

** correlation is significant at the 0.01 level (2-tailed).

**Table 4 ijerph-18-02031-t004:** Predictors for burden of prevention using ANOVA ^a^ test.

Model	Sum of Squares	df	Mean Square	F	Sig.
Regression	1370.866	6	228.478	5.824	0.000 ^b^
Residual	25930.344	661	39.229		
Total	27301.210	667			

df = degrees of freeddom; F value = the ratio of the mean regression sum of squares divided by the mean error sum of squares; Sig. = Threshold of significance/ Legend: ^a^ Dependent variable: burden of prevention; ^b^ Predictors: (constant), gender, type of practice, age, the region, chronic diseases of GP’s, seniority in the profession.

**Table 5 ijerph-18-02031-t005:** Predictors for symptoms of stress using ANOVA ^a^ test.

Model	Sum of Squares	df	Mean Square	F	Sig.
Regression	1416.461	6	236.077	6.029	0.000 ^b^
Residual	25,884.749	661	39.160		
Total	27,301.210	667			

^a^ Dependent variable: level of stress; ^b^ Predictors: (constant), seniority in the profession, type of practice, gender, the region, the work environment, age.

**Table 6 ijerph-18-02031-t006:** Predictors for adjustment to pandemic using ANOVA ^a^ test.

Model	Sum of Squares	df	Mean Square	F	Sig.
Regression	327.857	6	54.643	2.322	0.032 ^b^
Residual	15,558.017	661	23.537		
Total	15,885.874	667			

^a^ Dependent variable: degree of adjustment to pandemic; ^b^ Predictors: (constant), seniority in the profession, type of practice, gender, the region, the work environment, age.

## Data Availability

The data presented in this study are available on request from the corresponding author.

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
