# Peer review of "The Professional and Psycho-Emotional Impact of the COVID-19 Pandemic on Medical Care—A Romanian GPs’ Perspective"

_ijerph, 2021, doi:10.3390/ijerph18042031_

Round 1
Reviewer 1 Report
Reviewer report
Dear Author(s):
In accordance with the review of the article "The Professional and Psycho-Emotional Impact of the COVID -19 Pandemic on Medical Care - A Romanian GPs' Perspective" (ijerph-1119183), which has been assigned to my person, I communicate below what my decision was:
Accept after minor revision (corrections to minor methodological errors and text editing)).
Synopsis of the review
The only thing I can say in this regard is that the authors have scrupulously followed the suggestions, proposals for improvements and changes submitted by this reviewer. Furthermore, I consider that the final version of the article, as a result of my review and the other reviewers', is a significant improvement on the initial version, so I would be deeply flattered if this work is finally accepted.
I note that Table1 probably does not use the same font as the rest of the tables, so if this is the case, I suggest that you use a homogeneous font for the whole paper. One precision, lines [91-92] "Our neighboring countries, located near the Romanian border" I think it should be reworked: these countries are not "near" the Romanian Borders but they are exactly their bordering nations ... it is not that they are located near the Romanian border, it is that they delimit it. On the other hand, the phrase "Our neighboring countries": I do not consider it appropriate to use the first person plural in this paper but "Romania's bordering countries" or any analogous phrase.
In any case, I reaffirm the above, and suggest that this work be accepted.
With best wishes to the authors and the Romanian people,
The reviewer.
Author Response
Thank you for all the observations, I have modified the text, according to your specifications.
Best regards

Reviewer 2 Report
I kindly thank the Authors for having welcomed my suggestions. Now the Introduction and the Discussion sections appear more exhaustive and complete. The Materials and Methods section has been enriched with necessary information. The overall quality of the manuscript has been improved. I still think the Reference list is not as thorough as it should be, given the main topic of the study. However, I recommend the actual revised version of the manuscript to be published.
Author Response
Thank you, again, for all the observations, I have enriched the references, as you have reccomended.
Best regards

This manuscript is a resubmission of an earlier submission. The following is a list of the peer review reports and author responses from that submission.
Round 1
Reviewer 1 Report
The study is sincerely interesting, but the manuscript appears not completely exhaustive, and thus is not still ready for publication.
Please, see the attached file for detailed considerations.
I strongly recommend major revisions.

Reviewer 2 Report
Reviewer report
Dear Author(s):
In accordance with the review of the article "The Professional and Psycho-Emotional Impact of the COVID -19 Pandemic on Medical Care - A Romanian GPs' Perspective" (ijerph-1094155), which has been assigned to me, I communicate below what my decision was:
Reconsider after major revision (missing control in some experiments).
Synopsis of the review
In general terms the author(s) have done a good job, using a quite acceptable level of academic English, trying to present an eminently practical perspective on the Psycho-Emotional Impact of COVID-19 on the Romanian Society. These flaws are especially focused on the presentation of the paper, given that its content is really good: Once corrected this reviewer advises the publication of the paper.
I list them:
- Abstract: Please do not include numerical data of the results of your work: limit yourself to give a description of it and a brief recension of the achievements obtained in it. More specifically I refer to lines 20 and 21.
- Please use subscripts with "t" instead of t(675).
- In lines 40-45 you contextualize your study on the 45th week test rate in certain countries, which I think is a good idea: However, I consider it necessary that you mention all the countries bordering Romania, i.e. Bulgaria, Ukraine, Hungary (already mentioned), Serbia and Moldova.
- In line 92, there is probably a typographical error after Southwest (7.7%) with "and region". Please check if this is the case.
- In lines 89-91 you describe the geographic distribution of the analysis. Instead of doing this description "in text", do it with a graph: a pie chart or a bar chart. This will make your paper more readable for a future reader of your article.
- In order that a future reader unfamiliar with the geography and / or administrative divisions of Romania can quickly grasp the spatial location of the analysis, I consider it necessary that the authors state the equivalence of each of the 8 regions on which they have based their study, based on the nomenclature used by Eurostat (4 "Macroregiuni").
https://ec.europa.eu/eurostat/en/web/nuts/national-structures
- There is no need for explicit mention of the software and/or spreadsheet used.
- Final considerations.
Although it is true that there is no explicit obligation to include a review of the literature in your work, you do consider it necessary to place greater emphasis on other works already published that are more or less similar to yours. Evidently, the effects of the pandemic on the medical population were relatively similar in other countries, regardless of the level of resources of each nation. Keep this point in mind.
You have been able to find the "what" very efficiently, it would also be necessary for you to explain the "why". For example, I refer among others to the following points:
"Doctors in the North-East and North-West region have considered that the authorities were involved compared to those in Bucharest-Ilfov, who considered their involvement as insufficient"-> Okay, give an explanation on this point.
"Paradoxically, the Northwest is the most satisfied region in terms of the authorities' involvement compared to the Northeast, although the inhabitants of the latter region showed more symptoms of stress compared to their colleagues in the Northwest region. A somewhat paradoxical aspect is that physicians in the Northwest region, but also in the Southwest region, are more satisfied than those in the Southeast region"->Same, give an explanation in this regard.
Finally, I would like you to take into account the following consideration: There is not yet a real and contrasted worldwide homologation of COVID-19 cases, so making an effective comparison of the cases of one country with respect to another is not always a fruitful task. Please take this suggestion into account and include it as one of the limitations of your article.
Here you can find a very valuable quote:
https://www.europarl.europa.eu/doceo/document/E-9-2020-002553_EN.html
Once again I would like to insist to the authors who have done such a great work: I would be glad if this article could be published once the proposed suggestions were accomplished.
.
Best regards,
The reviewer